# A Comparative Study of Cancer Cells Susceptibility to Silver Nanoparticles Produced by Electron Beam

**DOI:** 10.3390/pharmaceutics15030962

**Published:** 2023-03-16

**Authors:** Evgenii V. Plotnikov, Maria S. Tretayakova, Diana Garibo-Ruíz, Ana G. Rodríguez-Hernández, Alexey N. Pestryakov, Yanis Toledano-Magaña, Nina Bogdanchikova

**Affiliations:** 1Research School of Chemistry & Applied Biomedical Sciences, Tomsk Polytechnic University, Tomsk 634050, Russia; 2Mental Health Research Institute, Tomsk National Research Medical Center, Russian Academy of Sciences, Tomsk 634014, Russia; 3Cancer Research Institute, Tomsk National Research Medical Center, Russian Academy of Sciences, Tomsk 634009, Russia; 4Nanoscience and Nanotechnology Center (CNyN), Campus Ensenada, National Autonomous University of Mexico (UNAM), Mexico City 04510, Mexico; 5Escuela de Ciencias de la Salud, Universidad Autónoma de Baja California, Ensenada 22890, Mexico

**Keywords:** electron beam, cytotoxicity, silver nanoparticles, tumor cells, anticancer agent

## Abstract

Introduction: Silver nanoparticles (AgNPs) have a wide range of bioactivity, which is highly dependent on particle size, shape, stabilizer, and production method. Here, we present the results of studies of AgNPs cytotoxic properties obtained by irradiation treatment of silver nitrate solution and various stabilizers by accelerating electron beam in a liquid medium. Methods: The results of studies of morphological characteristics of silver nanoparticles were obtained by transmission electron microscopy, UV-vis spectroscopy, and dynamic light scattering measurements. MTT test, alamar blue test, flow cytometry, and fluorescence microscopy were used to study the anti-cancer properties. As biological objects for standard tests, adhesive and suspension cell cultures of normal and tumor origin, including prostate cancer, ovarian cancer, breast cancer, colon cancer, neuroblastoma, and leukemia, were studied. Results: The results showed that the silver nanoparticles obtained by irradiation with polyvinylpyrrolidone and collagen hydrolysate are stable in solutions. Samples with different stabilizers were characterized by a wide average size distribution from 2 to 50 nm and low zeta potential from −7.3 to +12.4 mV. All AgNPs formulations showed a dose-dependent cytotoxic effect on tumor cells. It has been established that the particles obtained with the combination of polyvinylpyrrolidone/collagen hydrolysate have a relatively more pronounced cytotoxic effect in comparison to samples stabilized with only collagen or only polyvinylpyrrolidone. The minimum inhibitory concentrations for nanoparticles were less than 1 μg/mL for various types of tumor cells. It was found that neuroblastoma (SH-SY5Y) is the most susceptible, and ovarian cancer (SKOV-3) is the most resistant to the action of silver nanoparticles. The activity of the AgNPs formulation prepared with a mixture of PVP and PH studied in this work was higher that activity of other AgNPs formulations reported in the literature by about 50 times. Conclusions: The results indicate that the AgNPs formulations synthesized with an electron beam and stabilized with polyvinylpyrrolidone and protein hydrolysate deserve deep study for their further use in selective cancer treatment without harming healthy cells in the patient organism.

## 1. Introduction

Nanoparticles are widely used in various technology fields, but from a medical point of view, their potential is still far from being discovered. Currently, nanoparticles are used for the visualization of some molecular markers of diseases, diagnosis, malignant tumors treatment, and targeted delivery of drugs with controlled release and accumulation in tissues and organs. Nanoparticles are used as active components, for example, photosensitizers in photodynamic therapy of cancers or hyperthermic tumor destruction by heating nanoparticles [1]. However, the toxicity of nanoparticles for living organisms limits their medical use [2]. The biological properties of nanoparticles significantly depend on their size, shape, stabilizer type, and method of preparation [3,4]. In addition, particle nanosizing often leads to the appearance of new material properties or the enhancement of existing ones. However, this can also increase the potential hazard to human health [2]. The small sizes of nanoparticles (1–100 nm) allow them to penetrate through the epithelial and endothelial layers into the internal environment and body fluids, while migrating and being carried by the blood, penetrating even through dense histohematological barriers including the blood–brain barrier [5]. In this regard, the toxicity of nanoparticles is mainly realized through the following mechanisms: mechanical impact of nanoparticles and, in some cases, the formation of their aggregates with biological molecules; membrane integrity alteration and perforation; catalytic action of nanoparticles; enzymes damage and inhibition with cell metabolism disruption; deactivation of antioxidants and oxidative stress induced by nanoparticles; damage to cell cytoskeleton and internal organelles, primarily mitochondria; tissue inflammatory response and tissue damage due to immune response [6]. In many cases, toxicity is determined by the metal ions’ action during the dissolution of nanoparticles [7].

Among the wide range of nanoparticles, AgNPs occupy a special place due to their antibacterial, antifungal, antiviral, anticancer, etc., properties. Interest in AgNPs was mainly due to their outstanding antimicrobial activity, which allows them to be used in medicine and industry, where microflora suppression is required. Many works describe the antiviral, anti-inflammatory, antioxidant, and even hormetic stimulating effects of AgNPs [8,9,10,11,12,13].

Novel ways to use AgNPs in medicine continue to be sought; therefore, new risks appear. They are mainly associated with AgNPs toxic effects in human organisms [14]. Based on various data, it was confirmed that the biological properties of AgNPs can vary widely even when the particles have the same chemical composition [15]. 

The advancement of new methods of AgNPs preparation serves to develop nanoparticles with higher biomedical activity, higher stability, and lower toxicity [16,17,18,19]. Usually, various organic solvents and reducing agents are used for AgNPs production, and as a result, their traces could remain in the final AgNPs. These residual components are difficult to remove, which increases the toxicity of the obtained compositions based on AgNPs. In addition to the nanoparticles of the desired size, by-products of oxidized silver and its salts can also be formed, which also changes the bioactivity of the final product. One of the promising high-tech and waste-free methods for AgNPs fabrication is the reduction of silver ions by their exposure to an accelerated electron beam in an aqueous solution containing a stabilizer and silver nitrate [20]. Variations of this method have been developed, where the effect of the accelerated electron beam on the aqueous solution of silver nitrate was accompanied by adding the different stabilizers, including polyvinylpyrrolidone (PVP) [20] or protein hydrolysate (PH) [21]. This approach ensures the production of standardized AgNPs with high bioactivity and high stability in solution [22], which was used in biotechnology, medicine, veterinary, and agriculture [23,24,25]. At the same time, AgNPs synthesized with an electron beam and stabilized with PVP and PH showed low toxic effects towards hemolysis [26], and human lymphocytes [27]. The effect of AgNPs on tumor cells is of particular interest. The results obtained by our group for AgNPs synthesized with an electron beam and stabilized with PVP and PH allows us to consider AgNPs as potential agents for cancer therapy [28,29]. This study aimed to demonstrate that AgNPs formulations synthesized with an electron beam and stabilized with PVP and PH hydrolysate are promising agents as an alternative for cancer treatment. Characterization of AgNPs samples, cell growth inhibition, and cell death pathway were evaluated in SKOV-3, HCT-116, PC-3, SH-SY5Y, and Jurkat cell lines.

## 2. Materials and Methods

### 2.1. Synthesis of Silver Nanoparticles

Nanoparticles were obtained according to the methods described in patents [20,21]. Briefly, the method includes the following steps. First, a solution of collagen hydrolysate 18.8 wt.% (to obtain sample No. 1) or polyvinylpyrrolidone with a concentration of 18.8 wt.% (for samples No. 2 and No. 3). Then, silver nitrate solution necessary to reach 1.2% wt. of AgNPs (12 mg/mL of metallic silver) was prepared and stirred at room temperature until completely dissolved. The resulting silver salt solution was added to a vessel with the appropriate amount of stabilizer solution, intensively mixed, and exposed to an accelerated electron beam (voltage 30 kV) of high-energy (2–2.5 MeV) electrons with an absorbed dose of 15 kGy generated on a linear accelerator ILU-10 (Institute of Nuclear Physics, Novosibirsk, Russia). Electron beam treatment led to stable AgNPs formation. In general, the accelerated electrons have a relatively low damaging effect on organic polymers compared to gamma radiation [30]. For the comparative test of biological activity, all samples were diluted with distilled water. The samples for all tests were denominated as sample #1 (with collagen hydrolysate stabilizer), sample #2 (with polyvinylpyrrolidone stabilizer), and sample #3 (with a mixture of 70% of collagen hydrolysate and 30% of polyvinylpyrrolidone).

### 2.2. Characterization of the Silver Nanoparticles

#### 2.2.1. UV-Vis Spectroscopy

The optical properties of silver nanoparticles were characterized by measuring their absorption spectrum at the wavelength range from 200 to 800 nm at room temperature (25 °C) by UV−vis spectroscopy (Cary 60 UV-Vis Spectrophotometer, Agilent Technologies, Santa Clara, CA, USA). The absorption spectra of all samples were recorded for dilute aqueous solutions of the corresponding samples. Distilled water was used as a reference sample.

#### 2.2.2. Hydrodynamic Diameter and Zeta-Potential Analysis

AgNPs samples charge and hydrodynamic diameter distribution were determined by dynamic light scattering (Nano-ZS (Malvern Instruments Ltd., Malvern, UK)). The size distribution characteristics and Zeta-potential were measured in aqueous solutions at room temperature 25 °C with an equilibration time of 2 min. All samples were analyzed in triplicate.

#### 2.2.3. Transmission Electron Microscopy Analysis (TEM)

The transmission electron microscopy study was carried out on a JEM-2100F instrument (Jeol). A suspension was prepared based on ethanol and AgNPs lots and processed in an ultrasonic bath for 1 min. After that, the suspension was applied to a special copper mesh with a layer of formvar and a thin carbon film. Dried at room temperature for ~10 min.

### 2.3. Cytotoxicity Evaluation of Silver Nanoparticles

#### 2.3.1. Cell Cultures

The evaluation of the biological properties of the AgNPs was carried out on cell cultures in vitro. Standard tumor cell lines, including Jurkat (T-lymphoblastic leukemia), SH-SY5Y (neuroblastoma), HCT-116 (colon cancer), MCF-7 (breast cancer), MDA-MB-231 (breast cancer), SKOV-3 (ovarian cancer), and PC-3 (prostate cancer) (LLC “PrimeBioMed”, Russia”), were used. All the studied lines were brought into the phase of stable growth and, after 2–3 passages, were applied in the experiment. Adhesive cell line cultivation and subsequent cell experiments were performed using DMEM cell culture medium (Gibco, Billings, MT, USA) with GlutaMAX (cell supplement #35050061, Gibco, Billings, MT, USA), 10% FBS (fetal bovine serum, One Shot™ format, Brazil, Thermo Fisher Scientific, São Paulo, Brazil) and a mixture of antibiotics (penicillin/streptomycin mixture, Paneco, Moscow, Russia). Suspension cells were cultivated in RPMI 1640 medium with the same supplements. 

The preparation procedure included the following steps. Twenty-four hours before testing, 5000 cells of the corresponding cell line were seeded into each well of the 96-well plate and incubated for 24 h for cell adhesion and the start of cell growth and proliferation. After that, AgNPs with concentrations from 0.05 to 125 µg/mL (prepared by the serial dilution method) were added to the same plate. These concentrations refer to the concentration of metallic silver. The plate was incubated at 37 °C and under a 5% CO_2_ atmosphere for 24 h. Before starting the experiment, a visual check of morphological changes and living conditions of the cells was performed.

#### 2.3.2. Cytotoxicity Assay

To assess the cytotoxic effect of nanoparticles, an MTT test was performed. For the test, AgNPs with a final concentration of 0.05–125 µg/mL was added into a pre-seeded with cancer cells 96-well plate. Cells without exposure to AgNPs were used as a negative control. Cells incubated in a medium supplemented with 0.3% hydrogen peroxide were used as a positive control (dead cells). After 24 h of incubation, the medium from all wells was aspirated, and replaced with the fresh medium of the same composition containing 0.5 mg/mL MTT reagent (Paneco, Russia) was added and kept for 4 h at 37 °C in a CO_2_ incubator. The optical density was measured on a spectrophotometer (Multiskan FC, TermoFisher, Waltham, MA, USA) at a wavelength of 570 nm. The calculation was performed by subtracting the optical density of the background and the optical density of the positive (dead cells) control. Calculation of cell viability after exposure was performed as a percentage of alive cells in the experiment towards the viability control (cells without exposure to AgNPs). 

#### 2.3.3. Fluorescent Microscopy

Cells morphological changes assessment was performed by optical bright-field and fluorescent microscopy with differential staining according to the standard protocol (CalceinAM-Propidium iodide). For this, a stain solution was prepared with 0.5 μg/mL (CalceinAM) and 5 μg/mL (propidium iodide), and then it was added to the corresponding wells with cells. Incubation was carried out for 15 min at 37 °C, after which cultures were observed under a fluorescent microscope (AxioVert.A1, Zeiss, Jena, Germany).

#### 2.3.4. Flow Cytometry

The cytotoxic effect and the cell death pathway, viable, necrotic, and apoptotic cells at different phases were determined by flow cytometry. The cells were incubated with AgNPs for 24 h at 37 °C, and then they were washed and removed from the plate by exposure to trypsin solution. After that, they were precipitated by centrifugation (5 min, 200 g) and resuspended in a staining buffer containing a mixture of annexin V–FITC and propidium iodide. Next, live cells (negative staining), stained cells in the state of early apoptosis (annexin V-FITC-positive), late apoptosis (positive for both stains), and (necrotic) dead cells stained only with propidium iodide were counted by cytometer CytoFlex (Beckman Coulter, Brea, CA, USA) 

### 2.4. Statistical Analysis

The experiments were carried out in at least 6 replicates. The experimental results were statistically processed using the software GraphPad Prism 9 (GraphPad Software, Inc., San Diego, CA, USA). Results are presented as mean value with standard deviation. Differences between groups were considered significant at *p* < 0.05.

## 3. Results

### 3.1. Characterization of the Silver Nanoparticles

#### 3.1.1. Transmission Electron Microscopy (TEM) 

TEM is the main method for objective assessment of the morphology and size of nanoparticles. Micrographs of AgNPs samples showed that all samples contain detectable particles, located both in an isolated and grouped order (Figure 1). 

According to Figure 1, all samples showed that three studied samples have very similar particle size distribution, and they mainly consist of separated, spherical in shape, single particles with size 2–50 nm. Some of the nanoparticles have contact and produce aggregates with a size close to 100 nm.

#### 3.1.2. UV-Vis Spectroscopy

Optical characterization of the solution with the UV-visible spectrum is a simple method to confirm the presence of AgNPs. The absorption spectra of AgNPs samples obtained by electron beam irradiation are shown in Figure 2.

The spectrum of sample #1 has an absorption peak with a maximum at 426 nm. The spectrum of Sample #2 has an absorption band at 436 nm with a shoulder at 421 nm and a low intensive band at 520 nm (Figure 2). The spectrum of sample #3 has a band with a wide maximum in the interval of 420–460 nm. Peaks close to 400 nm are typical for AgNPs. The peak at 520 nm is attributed to large aggregated AgNPs [31,32]. 

These aggregates with a size of 100 nm were observed in TEM more frequently for sample 2 than for other samples (Figure 1). 

#### 3.1.3. Dynamic Light Scattering

The results of AgNPs hydrodynamic diameter distribution are shown in Figure 3A–C. As can be seen, there is some polydispersity in each sample, although the main fraction is always more than 60 percent. The peak between 1000 and 10,000 nm in Figure 3A indicates that in sample #1, the small part (9%) of nanoparticles was coagulated into microparticles. The data showed that the average hydrodynamic diameter of AgNPs samples was mainly in the range of 110–140 nm: 114.1 ± 0.340, 141.0 ± 0.189, and 142.6 ± 0.241 nm for samples #1, 2, and 3, respectively. The sample’s polydispersity index (PDI) ranged from 0.189 to 0.340, which indicates a relatively wide particle size distribution. 

The Zeta potential of the samples also varies significantly due to different stabilizers. The average zeta potentials were −7.3 ± 3.62, −3.8 ± 3.73, and +9.4 ± 6.28 mV for samples 1, 2, and 3, respectively. Usually, a large value of the zeta potential indicates the stability of the nanoparticles. When nanoparticles have a high charge, they repel, which increases their stability. For nanoparticles with zeta potential values less than ±25 mV, the aggregate formation increases, and the overall stability in suspension decreases [33]. In our case, surprisingly, despite the low charge (from −7.3 to +9.4 mV), after the electron beam, AgNPs remain stable probably due to the specific high protection by polyvinylpyrrolidone and collagen hydrolysate occurring at this method of sample preparation.

### 3.2. Cytotoxicity Properties of Silver Nanoparticles

The AgNPs obtained by electron beam irradiation were stable in water without any precipitation. AgNPs of samples #1–3 were added to the cell culture medium to obtain the desired concentration and used in cell biological activity tests. All samples were stable in cell media during experiments. 

#### 3.2.1. Cytotoxicity Assay

Figure 4 shows the significant cytotoxic effect of the studied AgNPs on all studied cancer cell lines. However, cytotoxicity differs significantly for samples with different stabilizers. The 3T3L1 line fibroblasts were the most resistant to silver nanoparticles. The tumor cell lines showed an AgNPs susceptibility higher than to 3T3L1 line fibroblasts (Figure 4). 

The overall resistance to AgNPs of tumor cell lines increases as follows: SH-SY5Y, MCF-7, Jurkat, MDA-231, HCT-116, PC-3, SKOV-3 (Figure 4). Thus, it was found that among the studied tumor cell lines, the most sensitive cell culture is neuroblastoma (SH-SY5Y), and the most resistant is ovarian carcinoma (SKOV-3). At the same time, it was found that AgNPs exhibit different cytotoxicity depending on the stabilizer used. All AgNPs samples contain the same concentration of silver and stabilizer, have very similar particle sizes measured with TEM and DLS, and differ only in the stabilizer. Half-maximal inhibitory concentrations for all studied tumor cells are compared for three investigated samples in Figure 5A. 

In addition, the selectivity index (SI) was calculated for each sample using the formula: SI = (IC50 for normal cell line 3T3L1)/(IC50 for each cancerous cell line) (Figure 5B). The samples efficacy against tumor cells is given by a SI > 1.0; in this case, all tumor cell lines present a SI > 1.0. 

The difference in cell viability for the most sensitive (SH-SY5Y, neuroblastoma) and the most resistant (SKOV-3, ovarian cancer) cell tumor lines reach 1 order magnitude (Figure 6A–C).

#### 3.2.2. Fluorescent Microscopy

The growth cell density and the number of viable and dead cells exposed to AgNPs in a wide range of concentrations were revealed by fluorescence microscopy (Figure 7). 

Figure 7A,B show for the most cytotoxic sample #3 live and dead cell densities for the most resistant tumor cell line SKOV-3 and the most susceptible SH-SY5Y neuroblastoma cells, respectively. The marked transition in cell viability is noted at concentrations 1.6 µg/mL for ovarian cancer and at concentrations 0.4 µg/mL for neuroblastoma. The rapid live-to-death transition within adjacent two-fold concentrations under the impact of AgNPs for all tested cell lines was observed. 

#### 3.2.3. Flow Cytometry

The results of flow cytometry testing are shown in Figure 8.

The cell suspension after AgNPs exposure was divided according to the viability status into the following fractions: live, early apoptosis, late apoptosis, and necrosis. Figure 8 illustrates that the main mechanism of cell death is apoptosis. The main fraction of cells after damage by AgNPs within 24 h starts the process of apoptosis, programmed cell death. In this experiment, an externally induced apoptosis (induced by AgNPs) pathway predominates. More than 90% of the cells are in a state of early and late apoptosis at concentrations of AgNPs above 1.6 μg/mL for all samples (Figure 8A–C). However, some necrotic cells (≤9%) were detected at high AgNPs concentrations. 

## 4. Discussion

Our results revealed that AgNPs samples evaluated have a selective and cytotoxic effect against cancer cell lines (Figure 1). Selectivity of AgNPs against tumor cell lines was shown by comparing the cytotoxic effect on non-cancer fibroblasts 3T3L1 line, which turned out to be up to 16 times more resistant to AgNPs samples than cancer cell lines. Furthermore, the three AgNPs samples evaluated have a SI > 1, demonstrating the need for further research regarding their use to treat cancer. 

In general, non-tumor cells are more resistant to the cytotoxic effect of silver. It is likely that tumor cell lines are more susceptible due to the increased cell replication rate [34,35,36]. Unlike ionic silver, nanoparticles could induce cell death primarily through mechanical impact and catalytic action enhancing lipid peroxidation, leading to proteotoxicity and necrotic cell death [35]. At the same time, AgNPs’ classical variants induced oxidative stress and apoptotic cell death which plays a significant role in these effects [37].

Surprisingly, even though neuroblastoma SH-SY5Y is characterized by not rapid growth with a doubling time of approximately 27 h [38], it was the most susceptible to AgNPs among the tumor cell lines used in the present study. This could be explained by the mechanism reported in which AgNPs induce endoplasmic reticulum stress and alter calcium metabolism, changing inositol phosphate function by the increased levels of phosphatase, which eventually leads to disrupted homeostasis in the mitochondria and apoptotic cell death [39]. Thus, the high degree of development of the protein-synthesizing apparatus of neuroblastoma cells makes them especially sensitive to such induced endoplasmic reticulum damage. 

Another non-specific mechanism of AgNPs on cells is oxidative stress. Even short-time exposure to AgNPs leads to reactive oxygen species production [37]. However, susceptibility to induced oxidative stress is highly variable among different types of tumor cell lines. This partly explains the different sensitivities found in our study. Summarizing the results, AgNPs cytotoxicity towards cancer cell lines decreases in the row of samples #3 > #1 > #2 (Figure 5). Sample #3, stabilized with a mixture of PVP and PH, showed a higher damaging effect compared to the studied tumor cells. However, the selectivity indexes are comparable for the three AgNPs samples except for MCF-7 and SH-SY5Y tumor cell lines. 

Moreover, the main mechanism of cell death upon exposure to three samples of AgNPs formulations is apoptosis (Figure 8). The contribution of primary necrotic cells increases only at high AgNPs concentrations (but even in this case, it does not exceed 9%). It indicates the increase in critical cancer cell damage under an excess of AgNPs, when the viability decreases so rapidly that the internal systems cannot respond adequately. The exposure to AgNPs is characterized by a very narrow critical concentration range between the state of cell death, partial viability, and normal cell culture growth, as shown in Figure 8. This pattern was detected for all studied cell lines.

In accordance with the reported in the literature, our results showed that tumor cell lines are more sensitive to AgNPs effect than normal 3T3L1 fibroblasts that exhibits noticeably greater resistance (Table 1). Thakore S. and co-workers reported a very low cytotoxicity of nanoparticles in relation to healthy cells, when the death of the fibroblast cell population did not exceed 30% at the maximum studied concentration 100 µg/mL [40]. The authors note that fibroblasts were significantly more resistant than A549 lung cancer cells.

Half-maximal inhibitory concentration for the studied samples on different tumor cell lines are in the range of 0.145–2.649 µg/mL, while the sensitivity of different cell types to the same AgNPs sample can differ up to 10 times (Figure 5). A significant scatter in the estimates of cytotoxicity is also observed according to the literature data (Table 1).

Analyzing the results for AgNPs formulations obtained by different methods, a huge difference in the CI50 for one cancer cell lines can be clearly seen (Table 1). All authors confirm that AgNPs are toxic to cancer cells and cause a decrease in cell viability. However, effective doses causing similar cytotoxic effects in some cases differ by an order of magnitude due to the properties differences in different AgNPs formulations. 

The results in Table 1 summarize the IC50 reported for diverse AgNPs, with the AgNPs sample #3 evaluated in this paper being the one that has the highest activity: 66, 205 times (SH-SY5Y); 4.4, 14.5 times (MDA-231); 2.4, 14.6, 23.4, 86, 226.8 times (MCF-7); 7.9, 44.6 times (HCT-116); 1.12, 3.25 times (PC-3); 130, 29.7 times (Jurkat); 7.7, 99.1 (SKOV-3); >57.8, 8.7–11.6 times (3T3L1). So, sample #3 is more active than AgNPs samples reported in the literature and is presented in Table 1 by an average of 52 times (5.220%). Only one AgNPs formulation showed very similar activity with sample #3, namely, the samples synthesized with green method [49] (Table 1), for PC-3 AgNPs in this work [49]. IC50 was just 12% higher than IC50 obtained in our work for sample #3. The IC50 for other AgNPs formulations of Table 1 were 2–200 times higher than IC50 of sample #3. The extremely high anticancer activity of sample #3 compared with other AgNPs formulations described in the literature (higher 52 times) and the selectivity index that shows it would be a safe formulation indicate its perspective and the necessity to continue the study of this formulation.

The comparison of our results with the literature data of Table 1 showed that Argovit AgNPs formulation is on average 52 times more active than the other sixteen AgNPs formulations presented in Table 1. AgNPs is not a molecule; it is very wide class of different formulations. Every formulation of AgNPs has different biological activity and toxicity, which depends on AgNPs size, shape, charge, stabilizer nature, method of preparation, impurities, etc. [56]. In our previous publication [57], it was shown that the activity of AgNPs formulations in relation to erythrocytes was very different, and Argovit AgNPs showed 40 times less toxicity (measured with AgNPs concentrations corresponding to 5% hemolysis) than the most toxic AgNPs formulation cited in [57]. The present work is another excellent illustration of great activity variation of different AgNPs formulations towards human cells. 

Some AgNPs formulations, described in Table 1, have very monodispersed AgNPs distributions, but all of them have lower activity than our AgNPs formulation. As indicated above, they were 2.4 to 250 times (240–2500%) less active than our formulation. Additionally, just one of them was only 12% less active. We think that the wide particle size distribution of our formulation is responsible for the fact that it has the highest activity for different cancer cell lines between seventeen various AgNPs formulations. These results allow us to hypothesize that if every cancer cell line needs a specific optimal AgNPs size, and if other formulations previously published and summarized in Table 1 do not have this specific size, then they will show low activity. In contrast, the wide size distribution of Argovit AgNPs formulation successfully provides the activity higher than activity of other AgNPs with monodispersed size, since at least part of AgNPs in this formulation has the optimized size for every specific cancer cell line that is provided by a wide particle size distribution. Obviously, this hypothesis needs further experimental verification. 

Optical spectra showed that the more monodispersed samples #1 and #2 have narrow peak at 420–440 nm, and sample #3 was characterized by the wider peak with a maximum at 420–460 nm, indicating the higher polydispersity of sample #3 than that of samples #1 and #2 (Figure 2). Thus, sample #3 presented in optical spectra the widest peak indicating the highest polydispersity showed the highest activity in cancer cell proliferation.

In the previous work of our group, it has been shown that AgNPs prepared with electron beam and stabilized with PVP do not have a noticeably damaging effect on primary cells cultures even at concentrations more than 100 times higher than inhibitory concentrations for tumor cell lines [28]. The authors noted that such nanoparticles showed 34.5 times greater activity against tumor cells than the well-known platinum-based cytostatic carboplatin. These results [28] directly indicate the selective nature of electron beam AgNPs formulation activity. Their significantly increased effect on tumors compared with healthy cells was obviously shown, and this creates certain prospects for their use as cytostatic agents. AgNPs reported in this paper, particularly sample #1 (stabilized with protein hydrolysate), showed an IC50 of 2.3 µg/mL on the highly aggressive human adenocarcinoma HCT-15, which is about 10 times more potent than carboplatin [29], while at a 260 times higher concentration (600 µg/mL), neither cytotoxic nor genotoxic damage was produced on human peripheral blood lymphocytes. Lymphocytes toxicity test is a sensitive, accurate, fast, and economical tool to evaluate whether materials are worthy of continued study of their effectiveness and toxicity for biomedical uses [27]. The main death pathway, elicited by sample #1 on HCT-15, was also apoptosis as it was determined for cancer cells here. Moreover, sample #1 acute oral toxicity on mice showed a lethal dose (LD50) of 2618 mg of Ag/Kg body weight determined in accordance with the OECD guideline 420 for Acute Oral Toxicity Assay, which classified it as practically nontoxic (Category 5) in accordance with the Globally Harmonized System of Classification and Labelling of Chemicals [29]. 

Thus, the aim set in the present work was achieved. All of the above results showed that the studied AgNPs formulations, prepared by electron beam and stabilized with PVP and protein hydrolysate, are characterized by high anticancer activity, low toxicity, and high stability, which are relevant characteristics for any pharmaceutical agents. These properties open new perspectives in the development of effective, selective, and safe AgNPs formulations for the cancer treatment and promise significant side effect reduction.

## 5. Conclusions

In this work, we present the cancer cell growth inhibition induced by three AgNPs formulations prepared by an accelerated electron beam of high-energy electrons which confers their unique biological properties. For the three studied AgNPs formulations, it was revealed that cancer cell inhibition by AgNPs was dose dependent, and the main mechanism of cell death was apoptosis. Moreover, the half-max inhibitory concentration for the seven studied cancer cell cultures varies by more than an order of magnitude, possibly due to different proliferation rates and tissue specificity of different tumor cell types. For all three studied AgNPs formulations, the sensitivity of seven cancer cell lines towards AgNPs decreases in the following order: SKOV-3 > PC-3 > HCT-116 > MDA-231 > Jurkat > MCF-7 > SH-SY5Y. The most sensitive to AgNPs were neuroblastic cells (SH-SY5Y), while the less sensitive ones were ovarian cancer cells (SKOV-3). The activity of the AgNPs formulation prepared with a mixture of PVP and PH, studied in this work, was about 50 times higher than the activity of other AgNPs formulations reported in the literature. 

The 3T3L1 fibroblasts cell line used as a control of normal cell (not cancer), was up to 16 times more resistant to AgNPs formulations than the tested tumor cell lines. These results, together with previous ones published by our group, indicate the selectivity of these AgNPs formulations, which are electron beam synthesized and stabilized with polivinilpirrolidone and protein hydrolysate, against cancer cells compared to healthy cells, with a selectivity index greater than 1 for all samples and tumor cell lines evaluated (SKOV-3, PC-3, HCT-116, MDA-231, Jurkat, MCF-7, and SH-SY5Y). The results indicate that these AgNPs formulations deserve deep study for their further use in selective cancer treatment without harming healthy cells in the patient organism.

## Figures and Tables

**Figure 1 pharmaceutics-15-00962-f001:**
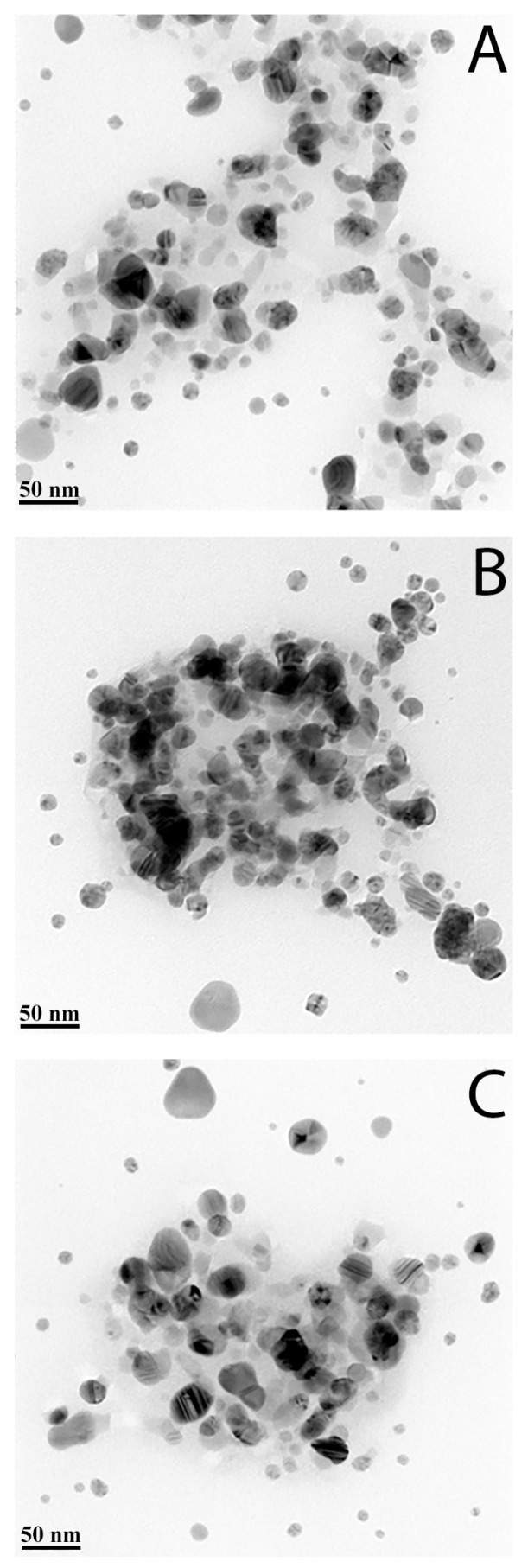
TEM images of AgNPs: (**A**)—sample #1; (**B**)—sample #2; (**C**)—sample #3.

**Figure 2 pharmaceutics-15-00962-f002:**
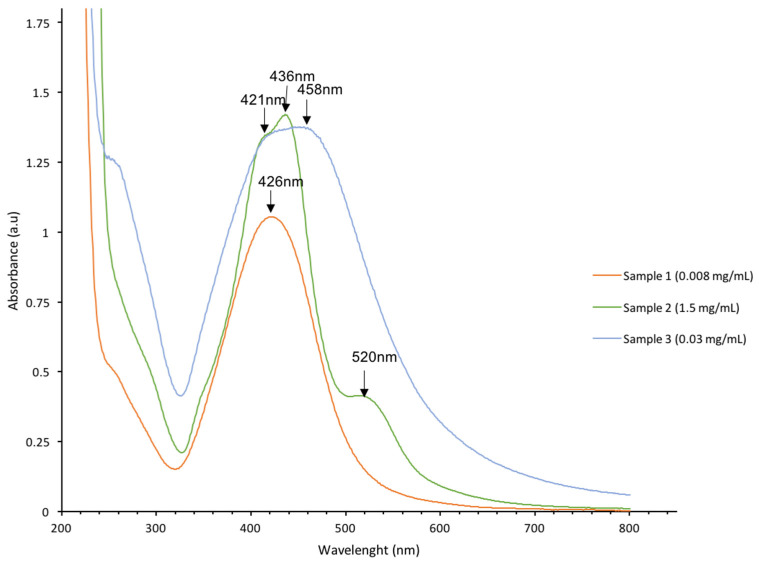
UV-visible spectra of AgNPs samples #1–3 with concentrations of: sample #1—0.008 mg/mL; sample #2—1.5 mg/mL; and sample #3—0.03 mg/mL.

**Figure 3 pharmaceutics-15-00962-f003:**
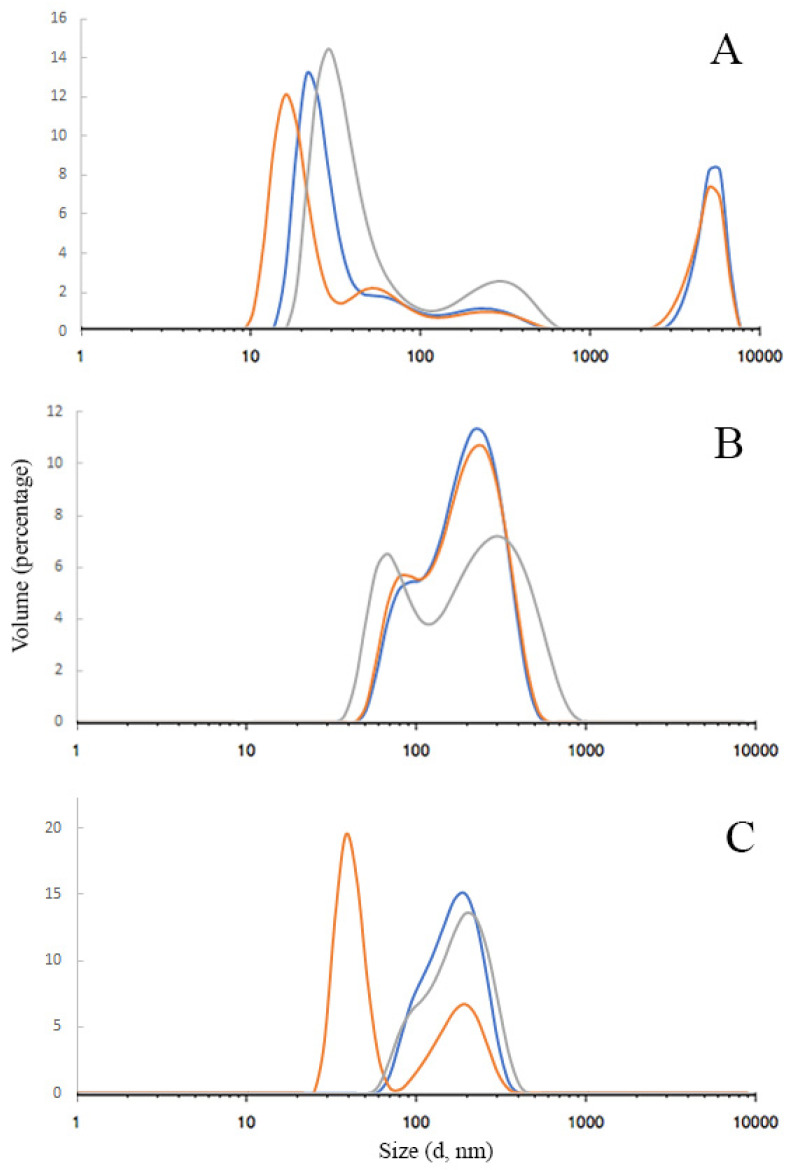
Silver nanoparticles size distribution measured by dynamic light scattering; (**A**)—sample #1 with collagen hydrolysate); (**B**)—sample #2 (with PVP); (**C**)—sample #3 (mixture). The different-colored curves represent the results obtained from three independent measurements.

**Figure 4 pharmaceutics-15-00962-f004:**
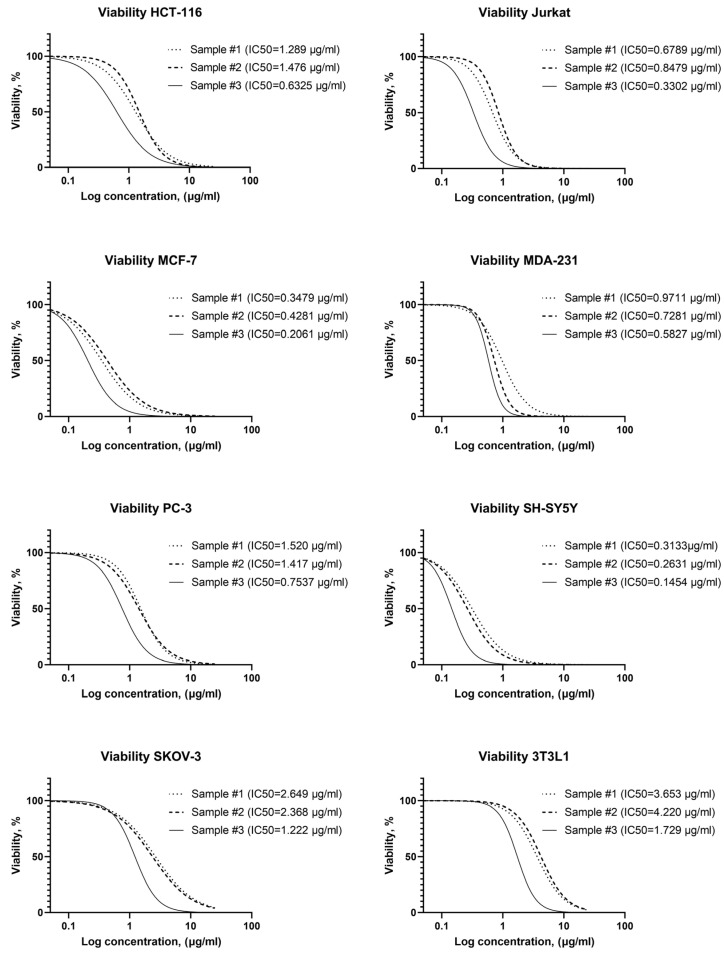
Dependence of cell viability vs. AgNPs concentrations with different stabilizers (AgNP samples #1–3).

**Figure 5 pharmaceutics-15-00962-f005:**
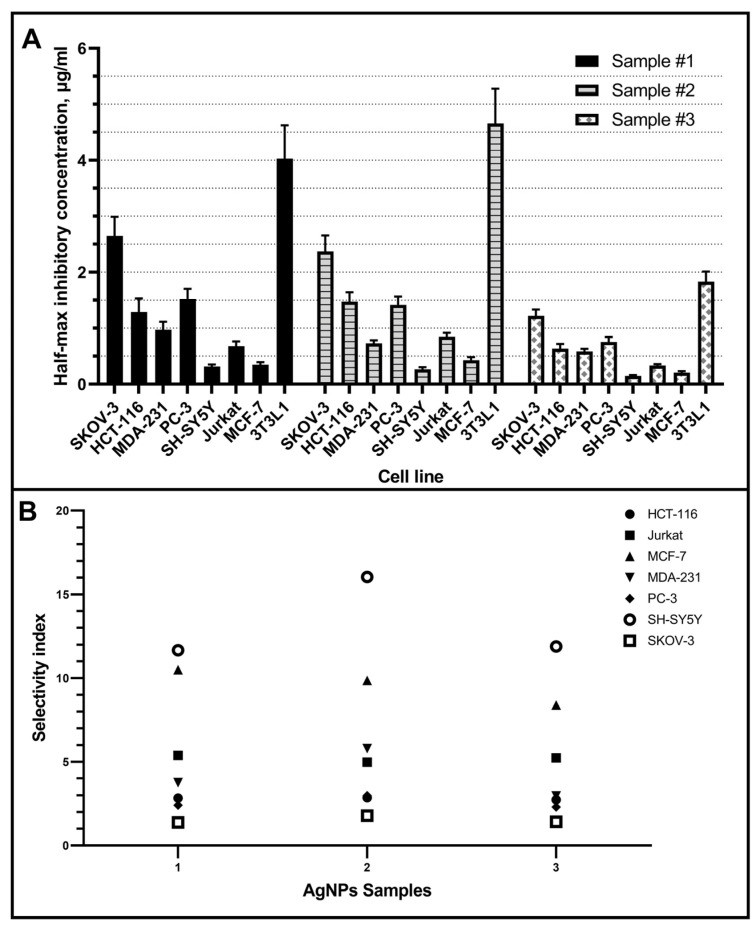
Half-maximal inhibitory concentration (**A**) and selectivity index (**B**) for the studied AgNPs samples on different tumor cell lines and fibroblasts 3T3L1 line as a control (bars are presented as CI 95%).

**Figure 6 pharmaceutics-15-00962-f006:**
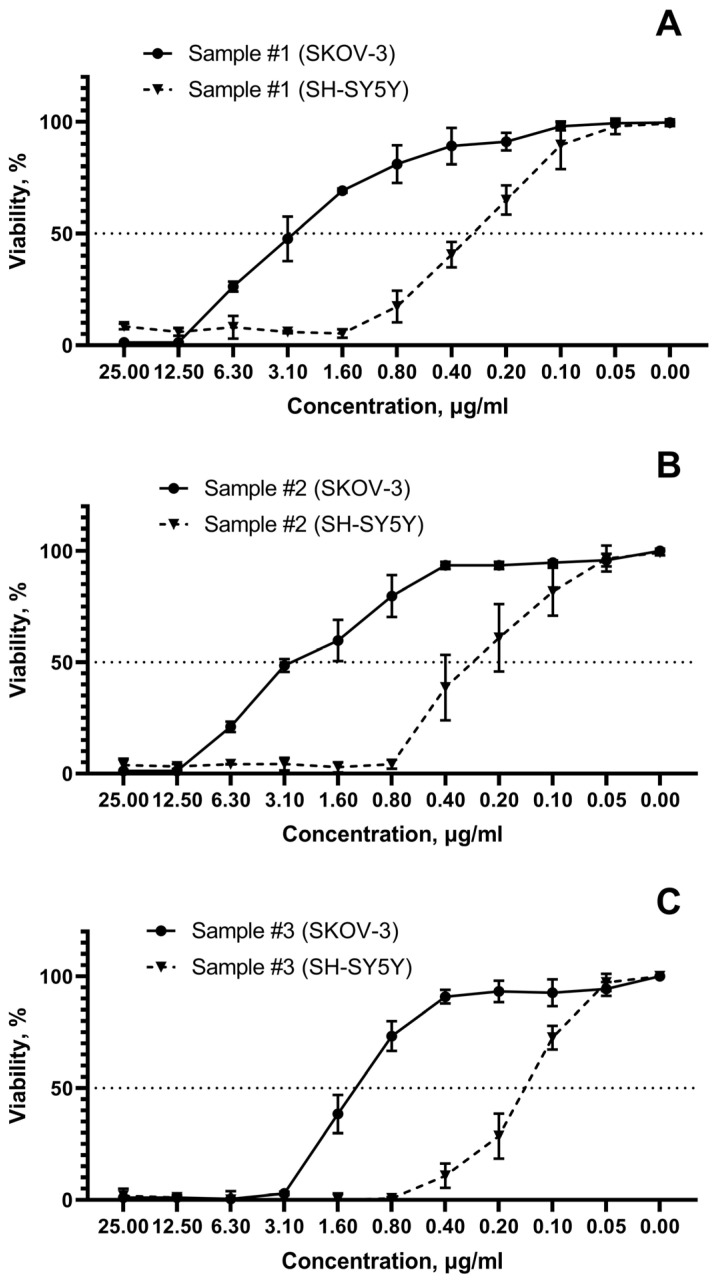
Cell viability depending on AgNPs concentration for the most susceptible (SH-SY5Y, neuroblastoma) and resistant (SKOV-3, ovarian cancer) tumor cell lines. (**A**)—sample #1; (**B**)—sample #2; (**C**)—sample #3.

**Figure 7 pharmaceutics-15-00962-f007:**
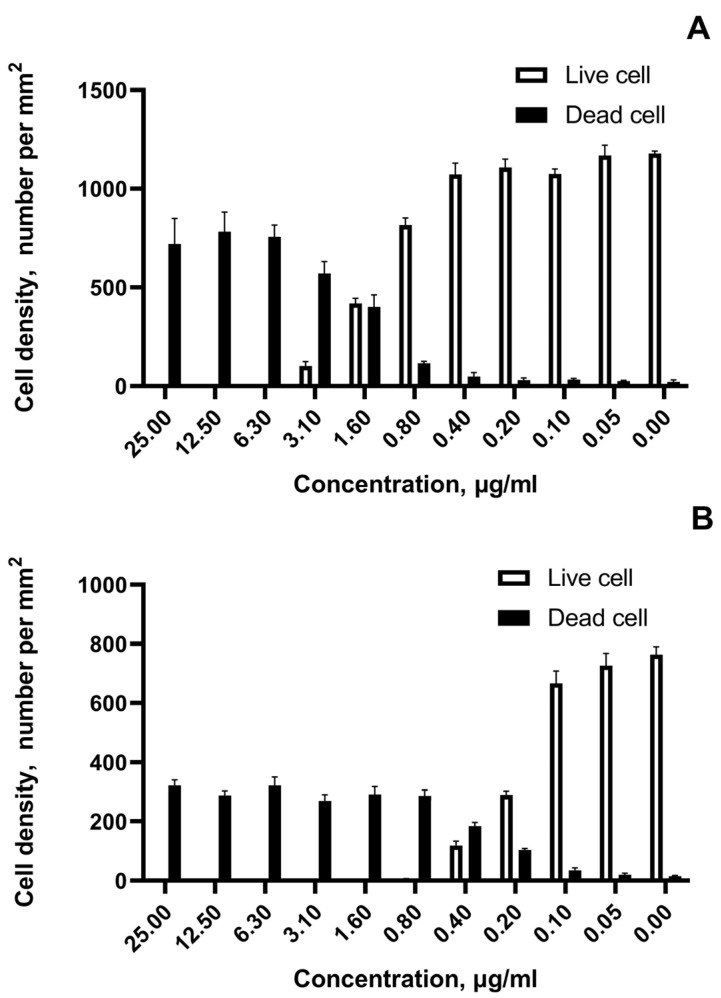
The growth cell density after exposition to the most cytotoxic sample of AgNPs (sample #3). (**A**)—number of viable and dying SKOV-3 cells per mm^2^; (**B**)—number of viable and dying SH-SY5Y cells per mm^2^.

**Figure 8 pharmaceutics-15-00962-f008:**
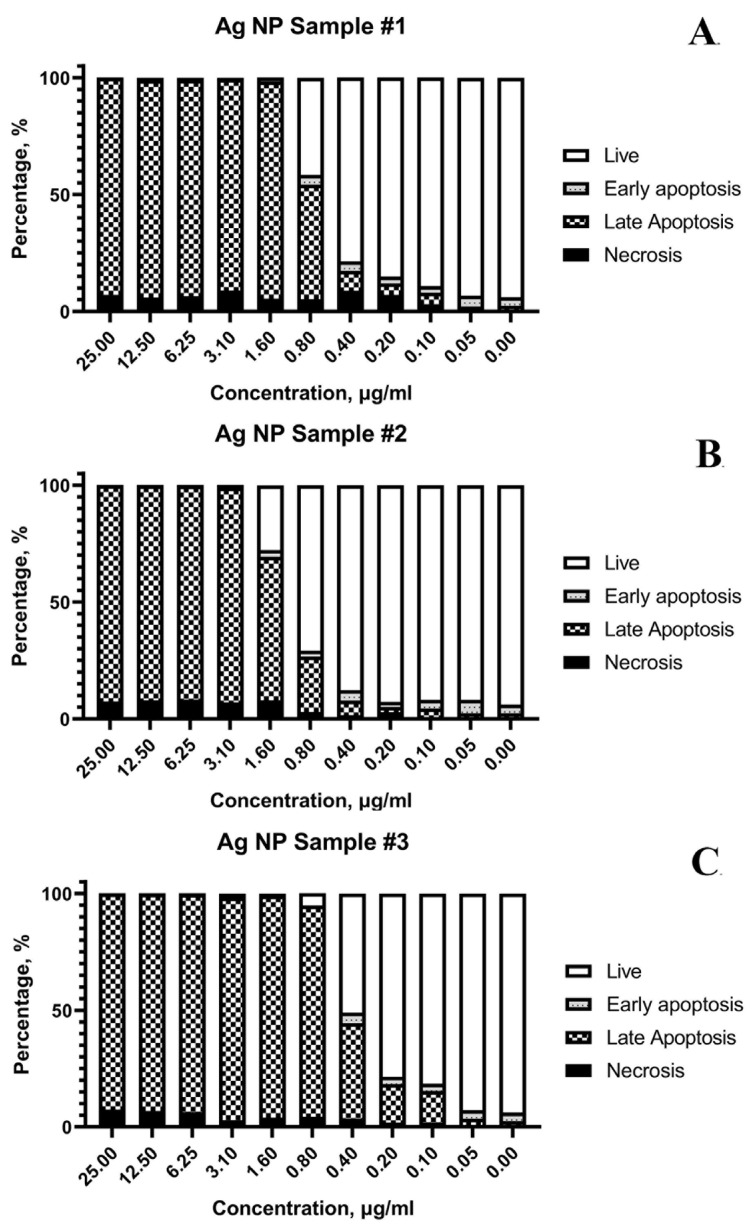
Distribution of Jurkat (T-lymphoblastic leukemia) cells by flow cytometry according to the variants of cell death after exposure to various AgNPs samples: (**A**)—sample #1; (**B**)—sample #2; (**C**)—sample #3.

**Table 1 pharmaceutics-15-00962-t001:** IC50 of cancer cell lines for different AgNPs formulations.

Preparation Method	Particle Size, nm	Stabilizer	Ag/Stabilizer Concentrations Ratio	Hydrodynamic Diameter, nm	Zeta Potential, mV	Cell Type	IC50, µg/mL	Reference
Commercial product (Colorobbia S.p.A., Vinci, Italy), series PARNASOS NAMA		Solutions were prepared by dissolving AgNPs in culture medium	AgNP 1% in water	20		SH-SY5Y	30.73 ± 3.20	[41]
Bio-reduction of silver nitrate	18			30		SH-SY5Y	10	[42]
Silver nitrate reduction by accelerated electron beam		Combined stabilizer PVP/protein hydrolysate	1.2/18.8 (wt.%)	142.6	+9.15	SH-SY5Y	0.15	This paper
Silver nitrate reduction by *B. funiculus* cultures supernatant.				20		MDA-MB-231	8.7	[43]
Commercial product (Argovit). Silver nitrate reduction by accelerated electron beam	35 ± 15	PVP	1.2/18.8 (wt.%)	70	−15	MDA-MB-231 MCF-7	2.62 ± 0.027 3.06 ± 0.014	[44]
Silver nitrate reduction by accelerated electron beam		Combined stabilizer PVP/protein hydrolysate	1.2/18.8 (wt.%)	142.6	+9.15	MDA-MB-231	0.6	This paper
Commercial product (Huzheng Nano Technology Limited Company (Shanghai, China)) 5, 20 and 50 nm.	5.9 ±3.3, 23.8 ± 6.7 47.5 ± 22.1	PVP				MCF-7	0.51 ± 0.02 14.33 ± 5.61 47.64 ± 14.67	[45]
Silver nitrate reduction by *P. fulgens* extracts	10 to 15 nm	*Potentilla fulgens* extract		39.04	−18 mV	MCF-7	4.91	[46]
Silver nitrate reduction by accelerated electron beam		Combined stabilizer PVP/protein hydrolysate	1.2/18.8 (wt.%)	142.6	+9.15	MCF-7	0.21	This paper
Silver nitrate reduction by flavonoid naringenin	6	naringenin (NAR)	NAR (50 µM) mixed with 2 mM AgNO_3_	6 ± 1		HCT-116	5	[47]
Silver nitrate thermal reduction by NaBH4		Trisodium citrate		57.4 ± 3.8	−39.4	HCT-116	28.11	[48]
Silver nitrate reduction by accelerated electron beam		Combined stabilizer PVP/protein hydrolysate	1.2/18.8 (wt.%)	142.6	+9.15	HCT-116	0.63	This paper
Silver nitrate reduction by polyphenolic fraction of flower extract *Cynara scolymus* L.		Polyphenolic fraction of flower extract *Cynara scolymus* L.		21.31 ± 0.431	−34.0 ± 4.45	PC-3	0.85 ± 0.01	[49]
Silver nitrate reduction by flower extract of *Cynara scolymus* L.		*Cynara scolymus* L. flower extract		26.57 ± 0.431	−29.9 ± 0.854	PC-3	2.47 ± 0.24	[50]
Silver nitrate reduction by accelerated electron beam		Combined stabilizer PVP/protein hydrolysate	1.2/18.8 (wt.%)	142.6	+9.15	PC-3	0.76	This paper
Silver nitrate with sodium borohydride using polyvinylpyrrolidone (PVP) as surface coating agent	67.1 ± 5.7	PVP		119.5 ± 1.4	−9.7 ± 0.2	Jurkat	42.9	[51]
Silver nitrate reduction by 1% trisodium citrate with 0.3% polyvinylpyrrolidone (PVP)	10–50	PVP				Jurkat	9.8	[52]
Silver nitrate reduction by accelerated electron beam		Combined stabilizer PVP/protein hydrolysate	1.2/18.8 (wt.%)	142.6	+9.15	Jurkat	0.33	This paper
Commercial product by nanoComposix. Powder. AgNPs with polyvinylpyrrolidone (PVP)	23.1 ± 6.9	PVP	Ag:PVP (15:85)	24.1 ± 0.4	−14.8 ± 0.5	SKOV-3	9.4 ± 1.4	[53]
Silver nitrate reduction by leaf extracts S. interrupta	5–14	Leaf extracts S. interrupta			− 28.9 mV	SKOV-3	120.87 ± 14.9	[54]
Silver nitrate reduction by accelerated electron beam		Combined stabilizer PVP/protein hydrolysate	1.2/18.8 (wt.%)	142.6	+9.15	SKOV-3	1.22	This paper
Microwave processing of a mixture sunflower oil and petroleum ether (1:1) and 0.01 M alcoholic silver nitrate solution	1–21			9	−27.31	3T3L1	At 100 µg/mL only 30% cell death	[40]
Silver nitrate reduction by myricetin	50 ± 5			55	−25.2 ± 0.1	3T3L1	15–20	[55]
Silver nitrate reduction by accelerated electron beam		Combined stabilizer PVP/protein hydrolysate	1.2/18.8 (wt.%)	142.6	+9.15	3T3L1	1.73	This paper

## Data Availability

Up on request.

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
