# Peer review of "A Comparative Study of Cancer Cells Susceptibility to Silver Nanoparticles Produced by Electron Beam"

_pharmaceutics, 2023, doi:10.3390/pharmaceutics15030962_

Round 1

Reviewer 1 Report

Dear authors,

Please consider the next suggestions to improve your manuscript:

The title of the manuscript 

COMPARATIVE STUDY OF THE CANCER CELLS SUSCEPTIBILITY WHEN EXPOSED TO SILVER NANOPARTICLES PRODUCED BY ELECTRON BEAM

 should be changed to:

 A COMPARATIVE STUDY OF CANCER CELLS SUSCEPTIBILITY TO SILVER NANOPARTICLES PRODUCED BY ELECTRON BEAM 

Authors must integrate all the figures by processing software as a single figure to avoid their fragmentation along the manuscript. 

 The quality of images must be increased. 

The interpretation of results must be improved, the explanation was very lack. 

Corrections: 

(line 13) The low sizes of nanoparticles allow them to penetrate through the epithelial and ….. 

Please define the small nanoparticle sizes (xxxx nm) ? 

(Line 90) Include the kV?

Author Response

Please, find an attached file

Reviewer 2 Report

It is an interesting study to compare silver nanoparticle (AgNP) toxicity in different cell lines. Besides the science part, the writing style and grammar need to be improved. 

The AgNPs produced in the study were very polydispersed from both TEM imaging and dynamic light scattering data. I wonder if the particles would aggregate in the cell culture medium. I suggest the authors investigate the stability of AgNPs in the cell culture up to 24 hours. It would be more convincing if the TEM images would have showed an evenly dispersed AgNP sample.

The authors have successfully demonstrated the IC50 differences between those three AgNP samples in different tumor cell lines. I also wonder to know the endocytosis of AgNPs in those different cell lines and if the tumor cells adopted different endocytic pathways which could cause different levels of IC50 for different AgNPs. It would be very interesting to see some cell imaging under light microscope or fluorescence microscope if AgNPs could be labeled with a fluorophore. 

The Figure 5A showed that the reactions of different tumor cell lines to the same AgNP sample have similar trend across all these three AgNP samples tested, which may indicate the mechanisms of cell death induced by AgNPs could be the same but the IC50 values could be different depending on the AgNP stabilized in different buffers. I suggest that the authors could be make some comments on the relationship between the AgNP property or stabilizer effects and the cell toxicity. 

Author Response

Please, find an attached file

Reviewer 3 Report

The manuscript presents synthesis of silver nanoparticles, study of their optical properties and cytotoxic effect. I find that the manuscript needs substantial editing, as there is a lack of connections between the part presenting the optical properties of the silver nanoparticles and their cancer cells susceptibility. I suggest that the authors pay attention to the following points in the text.

1. In the discussion, it should be clarified how the results for the optical absorption of the nanoparticles contribute to the explanation of cancer cells susceptibility.

2. The role of the nanoparticle’s sizes from the text it is not clear what the role of the size of the nanoparticles for their sensitivity of cancer cells or the effect observed is due to the different stabilizers used in the synthesis of NP’s.

3. How will you explain the difference in the description of the results of the Dynamic light scattering measurements, where the text claims "AgNPs samples was mainly in the range of 110–140 nm". While in Figure 3 no maximum is observed in the volume dispersion of the nanoparticles in this interval. What is the reason for the peak between 1000 and 10000 nm in fig. 3a. In Figure 3, the meaning of the different colored lines is not clear. Since the X-axis uses a logarithmic scale in figure 3, it is good to include minor ticks to X-axis.

Author Response

Please, find an attached file

Round 2

Reviewer 1 Report

Dear Authors, 

The comments and questions have been taken and revised version can be considered for publication.

Final Corrections: Please add visible scale bars in the Figure 1. TEM images.

Thanks.

Author Response

Thank you for your comments. We changed TEM images.

Reviewer 2 Report

The authors have addressed my comments properly. Other than that some English grammar mistakes need to be corrected, I would recommend publication of the manuscript.  

Author Response

Thank you for your recommendation. We improved our English.